# Optimization of Position and Number of Hotspot Detectors Using Artificial Neural Network and Genetic Algorithm to Estimate Material Levels Inside a Silo

**DOI:** 10.3390/s21134427

**Published:** 2021-06-28

**Authors:** Jeong Hoon Rhee, Sang Il Kim, Kang Min Lee, Moon Kyum Kim, Yun Mook Lim

**Affiliations:** 1School of Civil and Environmental Engineering, Yonsei University, 50 Yonsei-ro, Seodaemun-gu, Seoul 03722, Korea; oosi87@yonsei.ac.kr (J.H.R.); applymkk@yonsei.ac.kr (M.K.K.); 2ICE Meca Tech Co., Ltd., Sagimakgol-ro 45 Beon-gil 14, Jungwon-gu, Seongnam-si 13209, Korea; sikim@icemeca.com; 3Korea Midland Power Co., Ltd., 89-37, Boryeongbuk-ro 160, Boryeong-si 33439, Korea; champ1@komipo.co.kr

**Keywords:** artificial neural network, silo hotspot detector, material level, genetic algorithm, optimization

## Abstract

To realize efficient operation of a silo, level management of internal storage is crucial. In this study, to address the existing measurement limitations, a silo hotspot detector, which is typically utilized for internal silo temperature monitoring, was employed. The internal temperature data measured using the hotspot detectors were used to train an artificial neural network (ANN) algorithm to predict the level of the internal storage of the silo. The prediction accuracy was evaluated by comparing the predicted data with ground truth data. We combined the ANN model with the genetic algorithm (GA) to improve the prediction accuracy and establish efficient sensor installation positions and number to proceed with optimization. Simulation results demonstrated that the best predictive performance (up to 97% accuracy) was achieved when the ANN structure was 9-19-19-1. Furthermore, the numbers of efficient sensors and sensors positions determined using the proposed ANN-GA technique were reduced from seven to five or four, thereby ensuring economic feasibility.

## 1. Introduction

Coal and biomass are important sources of energy in the modern society. Industries that generally utilize these resources synthesize, collect, and store them in large quantities for utilization as per the requirement. Therefore, safe storage is a crucial aspect. Silos are employed in several industries as the most preferred structure for storing resources because of their storage capacity and multiple control functions. Therefore, determining the level of resources in the silo is critical for resource management and silo operations. Several tools and methods have been used for this purpose. The most commonly used methods include the plumb-bob method, weight and cable method, ultrasonic techniques, guided wave radar (GWR), laser meters, load cells, and thru-air radar (TAR) [1,2,3]. Each technique has its own advantages and disadvantages. Ultrasonic and laser techniques are highly effective for measuring the material level inside a silo. However, they are not practical owing to the presence of dust inside the silo [2,4]. The application of a load cell to measure the amount of material inside a silo is another widely used simple method, but this method cannot be applied to large silos. However, GWR and TAR techniques can address this disadvantage. Although GWR systems are generally more precise, they require an induction cell capable of transmitting and receiving electromagnetic (EM) energy while performing measurements at a single level point [2,5]. TAR requires an antenna for transmission and reception of the EM waves. Furthermore, the beam width used in this technique is limited. The number of antennas to be installed and the required measurement points increases with the increase in silo diameter. Consequently, a more economical and practical method is required to measure the material level inside a silo.

In this study, we employed a silo hotspot detector to overcome these limitations. This hotspot detector is developed to detect the possibility of fire, such as natural ignition of the silo’s internal storage. It is installed inside the silo and detects the possibility of fire by directly measuring the internal and external temperatures of the storage in real time. After checking the temperature measurement results of the detector, it can be seen that the temperatures inside and outside the material are different. This can be used to infer the internal material level of the silo based on the temperature results of the detector. In some cases, the error value is small compared to the actual value; however, it is also confirmed that significant errors occur. Therefore, it is difficult to estimate accurate level values simply by dividing the boundaries; because of this, other methods are needed to mitigate this problem.

Machine learning (ML) techniques, which are applied in various ways in various fields, can be used to solve the aforementioned problems. ML techniques analyze the existing data to predict other relevant data and future results [6]. The most commonly used ML techniques include the k-nearest neighbor (KNN) algorithm [7], support vector machine (SVM) [8], and artificial neural networks (ANNs) [8]. ANNs are generally preferred owing to their simplified mathematical definitions and short training times. A back-propagation neural network (BPNN), which is a specific type of ANN, is a multi-layer feed-forward neural network (MFNN) trained using a back-propagation (BP) algorithm. The BP algorithm is a local search algorithm that iteratively updates the weight and bias of neural networks to minimize errors with real data and ANN prediction results. BPNN has been utilized by many researchers because of its ability to map relationships between two data in complex, nonlinear relationships.

Furthermore, genetic algorithms (GAs) are optimization methods that use selection, crossover, and mutation operators to search the response space more efficiently without multi-pointing and initial conjecture in each iteration. Chandwani et al. [9] studied the effectiveness of combining ANNs and GAs for slump modeling of ready-mixed concrete based on cement, fly ash, sand, coarse aggregate, mixing, and water-binder ratios. Gao et al. [10] developed a fault-finding system for lithium-ion batteries using GA optimization to predict the optimal operating parameters. Cui et al. [11] utilized a GA to optimize the indoor Wi-Fi positioning. Zhang et al. [12] conducted a GA-ANN-based short-term wind speed prediction study for wind generator installation. Li et al. [13] optimized the internal combustion engine strategy through the ANN model, and an ANN-GA model was introduced to improve accuracy and stability. In addition, there are a variety of studies that combine neural networks (NNs) and GAs to solve the problem [14,15,16]. As such, research using ANNs and GAs is still actively applied to the present in various engineering domains. Nevertheless, methodologies using ANNs and GA to measure the internal material level of silos have not been studied so far.

To overcome the limitations of existing silo material-level measurement techniques, this study proposes a novel method for measuring material level by applying data obtained through a silo hotspot detector, which monitors silo internal temperature, to ANN. In addition, we propose a method to combine GA and ANN to obtain the optimal number of sensors and installation position to improve accuracy and secure economic feasibility in real-world applications. The remainder of this paper is organized as follows: Section 2 introduces the silo hotspot detector and the experimental process. Section 3 discusses the applied ML algorithm, the application method, and the analysis process. Section 4 discusses the GA, the algorithm applied for optimization, and describes the optimization process. The results and discussion are presented in Section 5 and Section 6, respectively. The final section concludes the study and discusses the scope of future research.

## 2. Dataset Deployment

### 2.1. Experimental Data Collection

As mentioned above, the data required for this study were obtained from the experiments. The silo hotspot detector used in this study is shown in Figure 1, and its specifications are listed in Table 1. To obtain the temperature data measured during the silo operation, the detector was installed, as shown in Figure 2. The specifications of the target silo and positions of the detector and sensors are shown in Figure 2. The temperature was measured at seven positions by installing the temperature sensors, as shown in Figure 2. The silo internal material level, which is another data required for this study, was measured using the well-known plumb-bob method. The machines and measurement methods used in the plumb-bob method are shown in Figure 3. It is a device that can send a rope-type machine with ironing tips down from the top of the silo to measure the distance lowered when it touches the surface of the material and estimates the level of the current silo material. The specifications for the level-measuring devices are shown in Table 2.

### 2.2. Experimental Result

Some experimental results, which were obtained from the prediction of the level of materials inside the silo, were extracted and analyzed. Figure 4 shows the experimental results for 10 cases. The temperature data measured through the hotspot detector and the section with a large temperature difference in each case were assumed to be at the material level. Table 3 shows a comparison between the assumed material-level values and the measured values. This indicates that some level values can be estimated; however, there are instances where large differences between the assumed and measured values can be seen. Therefore, it can be confirmed that other methods are needed to obtain reliable material-level values.

### 2.3. Feature Extraction

First, from the acquired data, the features used for learning were screened for efficient learning and for improving the accuracy of the ML algorithm; these features included the following three data types:

Temperature data from the sensors (F_1_): The data recorded by the seven temperature sensors installed on the silo hotspot detector, which could be located above or below the material level, were included in one dataset for each level and represented the principal data in this study.Atmospheric temperature (F_2_): This refers to the ambient air temperature around the silo, which acts as a reference for determining the air temperature inside the silo.Number of sensors below the material level (F_3_): The seven temperature sensors in the silo hotspot detector could be located above or below the material level. Therefore, it is important to account for the number of internal sensors for each set of acquired data in the dataset. The number of internal sensors at different material levels is shown in Table 4.

### 2.4. Dataset Implementation

To ensure efficient training and accuracy prediction by the ANN model, the datasets were constrained within the range of the training data. The extreme (maximum and minimum) values of the training dataset were set such that they included the values in all the datasets used [9]. The training and test datasets comprising the temperature and silo-level values are listed in Table 5. A total of 3080 datasets were utilized as training data and test data.

## 3. Methodology

In this study, the neural network toolbox included in the commercial software MATLAB R2020a was used to perform the analysis.

### 3.1. Artificial Neural Network (ANN)

ANN is an information-processing paradigm that processes complex systems by mimicking the learning processes of the human brain. Although the input data and output data relationships are difficult to define, application of ANN allows adequate correlation between two, e.g., complex nonlinear relationships. The accuracy of the prediction results obtained using ANNs is considerably influenced by the ANN architecture, which includes the number of layers in the network, number of neurons in each layer, type of transmission, and training functions in each layer [17] and different architectures that need to be applied for each problem. The architecture applicable to this study is discussed in Section 3.2.

In this study, we applied the MFNN, a type of MLP [18,19,20,21,22,23], which has proven to be well-suited to several applications. This neural network comprises three layers: an input layer, a hidden layer, and an output layer, as shown in Figure 5.

The first step for applying ANN to the problem to be solved is constructing a training dataset for the problem. The training data are propagated from the input layer to the output layer through the hidden layer, where the output value is generated after adjusting the weights and biases of each layer. The error between the predicted output and actual value is calculated and propagated in the reverse direction. The algorithm re-adjusts the weights and biases according to the calculated error. After this process is repeated, the final prediction results are determined when the target error value is reached.

For the efficient learning of ANNs, appropriate values for the learning rate and momentum parameter were selected, as shown in Table 6. The learning rate is a common parameter in numerous learning algorithms, affecting the rate at which ANNs obtain solutions. In the ANN, the learning rate was similar to the step-size parameter of the gradient descent algorithm. If the step size is too large, the system will go back and forth for the true solution, or it will be complete. If the step size is too small, the system will require a significant amount of time to converge to the final solution. The momentum parameter prevents the system from focusing on the local minimum or saddle point. The high momentum parameters increase the convergence rate of the system but can overshoot the minimum value; this can destabilize the system. In contrast, low momentum parameters are unavoidable local minima and slow down the training speed of the system [24]. The combination of these parameters assists BP algorithms to overcome the local minima effect. The weight change is determined by learning rate and the momentum parameter, such as Equations (1) and (2).
(1)$$\Delta w_{n}=\alpha \Delta w_{n-1}-\eta \frac{\partial e}{\partial w}$$
where
(2)$$e=\frac{1}{N}\overset{N}{\underset{i=1}{\sum }}{\left(T_{i}-K_{i}\right)}^{2}$$

Here, w represents the weight between two neurons; Δw and $\Delta w_{n-1}$ represent the changes in weights at n and n−1 iterations, respectively, α is the kinetic factor, η is the learning rate, e is the computational error, $T_{i}$ is the actual output, and $K_{i}$ is the prediction output.

### 3.2. Setup of ANN Architecture and Parameters

#### 3.2.1. Number of Hidden Layers

Many researchers have reported that an ANN with a three-layer structure comprising one hidden layer functions appropriately [18,21,23,25,26,27,28]. However, other researchers have suggested using two hidden layers to analyze the nonlinear relationship between the data [29]; moreover, in some studies, two hidden layers were reported as optimal structures for problem solving. [19,20,30,31]. Optimization rules or concrete methods for determining the number of hidden layers are yet to be established. Consequently, in this study, we compared the network features with one and two hidden layers using trial and error methods and determined the optimal number of hidden layers.

#### 3.2.2. Number of Neurons in the Layers

The number of neurons in the input and output layers is typically determined at the beginning of the study, as it determines the actual number of data used and the number predicted. Because no optimization rule is available to determine the required number of neurons in the hidden layer, we used the values proposed by Zhang et al. [32] for this purpose, that is, n/2, n, 2n, and 2n + 1, where *n* represents the number of input variables. We selected the optimal number of neurons by comparing the results for each case. Because the number of neurons in the input layer was nine, the ANN structures in this study were 9-5-1, 9-9-1, 9-18-1, 9-19-1, 9-5-5-1, 9-9-9-1, 9-18-18-1, and 9-19-19-1.

#### 3.2.3. Transfer and Training Function

In an ANN, the transfer function determines the strength of the output value after converting the weighted sum of the input data to the output neuron. A standard criteria for determining the transfer function has not yet been established; hence, the function should be adapted to the problem that is being solved [33]. Therefore, in this study, the following commonly used transfer functions were adopted: the tangent transfer function (tansig) for hidden layers and the linear transfer function (purelin) for the output layers [34]. The training function employed a scaled conjugate gradient (SCG) algorithm.

A systematic update of the weight and deflection values was performed using the SCG developed by Møller [35]. This algorithm combines a model-trust region approach with a conjugate gradient approach; the latter uses a step-size scaling mechanism to avoid time-consuming line searches per learning iteration. Moreover, the SCG algorithm does not consider user-dependent parameters with important values [35]. The overall flow diagram of the ANN models applied in this study is shown in Figure 6; the parameters of the architecture are listed in Table 6.

### 3.3. Evaluation of the Trained Model Performance

Two statistical parameters were adopted to evaluate the performance of the trained ANN model. These included the mean absolute error (MAE), mean squared error (MSE), and coefficient of correlation (R). The performance metrics are calculated as follows:(3)$$\mathrm{MAE}=\frac{1}{N}\overset{N}{\underset{i=1}{\sum }}\left|T_{i}-K_{i}\right|$$
(4)$$\mathrm{MSE}=\frac{1}{N}\overset{N}{\underset{i=1}{\sum }}{\left(T_{i}-K_{i}\right)}^{2}$$
(5)$$R=\left(\frac{\sum_{i=1}^{N}\left(\left(T_{i}-\bar{T}\right)\left(K_{i}-\bar{K}\right)\right)}{\sqrt{\sum_{i=1}^{N}{\left(T_{i}-\bar{T}\right)}^{2}\sum_{i=1}^{N}{\left(K_{i}-\bar{K}\right)}^{2}}}\right)$$
where *N* is the total number of dataset, T is the target observed value, K is the ANN predicted output value, $\bar{T}$ is the average of the observed value, and $\bar{K}$ is the average of the ANN predicted value. MAE represents the mean value converted from the difference between the observed and the predicted value to the absolute value. MSE represents the square root of the mean error between the observed and predicted values. A low MAE and MSE indicates better predictive performance. The association strength between observations and predictions was quantified using Pearson’s correlation coefficient (R). However, R depends on the linear relationship between the real and predicted values, resulting in biased results if the relationship between the two values contains outliers. Therefore, the closer the R value is to 1, the more accurate the predicted results from the ANN model. Combining the three aforementioned performance metrics can provide unbiased estimates of the predictive capabilities of neural network models.

## 4. Optimization

In this study, the silo internal material level was predicted using temperature data and ANN; additionally, optimization tasks were performed to improve the prediction accuracy, minimize the number of temperature measurement points, and optimize positions. The subsequent sections describe the algorithms and methods applied to the performed optimization tasks.

### 4.1. Genetic Algorithm (GA)

GA is an optimization algorithm that mimics the principles of natural evolution of living things and is widely applied and utilized in various domains, including engineering [36]. Figure 7 illustrates the concept of GA. The algorithm leverages the selection, crossing, and mutation operators shown in the biological evolution process to produce better results in the next generation compared to the present generation. The selection operator evaluates the fit of the existing response elements and selects the best one among them. The crossover operator combines the two selected responses from the parent to create a new, unique response. Finally, the mutation operator changes the elements of a given response, creating a wider variety of responses. Responses that evaluate the suitability of new responses created by crossover and mutation operators and exhibit a high fit are passed on to the next generation. This process is repeated until an optimal fitness score is obtained for a problem.

This allows GA to navigate solutions by mutual cooperation between multiple objects, making it easier to find better solutions compared to exploration of simple parallel solutions. Moreover, unlike other algorithms that require differential values of the evaluation function, GA only requires that the current adaptation is discernible. Therefore, the algorithm is simple and can be applied even when the evaluation function is discontinuous. For this reason, this study performed optimization tasks using GA.

### 4.2. Optimizing Position and Number of Temperature Measurement Points

In this study, the objective of optimizing the silo internal material-level prediction process was to improve the accuracy of the material-level prediction while determining the optimal position and number of temperature measurement points. To this end, we used the ANN-GA hybrid approach. The steps involved in the ANN-GA approach are depicted in Figure 8. The parameters used are listed in Table 7.

Steps involved in the ANN-GA process:Selection of the number of temperature measurement positions (*ts*) and configuring the ANN architecture. Establishing training algorithms and termination conditions; optimizing network trainingSelection of parameters, that is, the population size (*N_pop_*) and the maximum number of generations (*G*_en_), for the optimization process.Determining the output value of each response using trained ANNs; determining the suitability of the output value using the objective function.Evaluation of conformity values using selection, crossover, and mutation.Generation of a new response from the previous step; evaluation of the conformity value of the new response. Determining the ranking of all responses based on the target function value with the most appropriate *N_pop_* value forming the next generation.Repetition of steps 3–5 until one of the following termination conditions is satisfied:The maximum number of generations is exceeded.The fitness function reaches the target value or a specific deviation from the target value.There is no improvement in the fitness value during the specified generation.

## 5. Results and Discussion

### 5.1. ANN Model Sensitivity Based on Input Value Combination

Prior to the silo material level prediction over ANN, we confirmed the sensitivity of the ANN model when multiple combinations of data features compiled in Section 2.3 were applied as input data. The total data combinations are F_1_ (Case 1), F_1_ + F_2_ (Case 2), F_1_ + F_3_ (Case 3), and F_1_ + F_2_ + F_3_ (Case 4). The results for each case are shown in Table 8. Sensitivity analysis confirms that Case 4 performs best. Therefore, in this analysis, the combination of input data is applied as shown in Case 4.

### 5.2. Training Results

The training results are summarized in Figure 9 and Table 9. For one hidden layer, the MSE value was minimum at 0.0786, and the R value was maximum at 0.9912 in Case 4. For two hidden layers, the MAE value was minimum at 0.2377, the MSE value was minimum at 0.0315, and the R value was maximum at 0.99612 in Case 8. As shown in Figure 9, the ANN architecture had an identical impact on the MAE, MSE, and R values. The prediction accuracy was higher when the number of hidden layers was two than in one, and the prediction accuracy increased as the number of neurons increased.

In general, the accuracy of ANN training results vary with the number of hidden layers and neurons [17]. This effect was identified in the present study. It was confirmed that the more suitable the structure was for the problem it sought to solve, the more accurate the ANN training result was.

### 5.3. Test Results

Tests were performed on eight ANN models utilizing data not used for training. Each structure consists of a different number of hidden layers and neurons. The analysis of test results allowed us to determine whether overfitting may occur during the training process. This is an important aspect when applying the developed ANN model to real-world sites.

The test results are shown in Figure 10 and Table 9. For one hidden layer, the minimum value of MAE and MSE was 0.2709 and 0.1808 in Case 4, and the maximum value of R was 0.9758 in Case 1. For two hidden layers, the MAE value was minimum (0.2391) in Case 8, the MSE value was minimum (0.1545), and the R value was maximum (0.98406) in Case 6. The test results differed from those of the training results. The MAE and MSE value was higher than the training result, and the R value was lower. This indicates that the test results were less accurate than the training results. However, despite these differences, the test results showed high accuracy and confirmed that no overfitting occurred during the training process.

To verify the accuracy of the test results more directly, the actual and predictive results were compared, and the differences were identified. Considering the training and test results for various ANN architectures, we compared the test results for the selected structure (Case 8: 9-19-19-1) with the actual results, as shown in Figure 11. The comparison results confirm the following results.

The ANN predicted changes in the level of materials inside the silo owing to material discharge and charging.An inspection of the results confirmed that the temperature data could be used to predict the level of material inside the silo.

### 5.4. Optimization Results

As noted in Section 4.2, the variable in the optimization process indicates the position of the internal silo temperature measurement point. Optimization was performed to obtain the optimal measurement position for cases with three, four, five, and six optimization points. Based on the target function (MSE), we evaluated 10 *ts* populations per generation. As an end condition, we set the optimization process to terminate when the fitness value did not change upon repeating more than 150 generations. Furthermore, ANN training was conducted using the optimized position, and the material level was predicted. Figure 12 shows the optimization endpoints for each variable.

Figure 13 shows the plot of the optimized position based on the number of temperature measurement points inside the silo. Table 10 shows the ANN training results when the sensors were placed at optimized positions based on the number of internal silo temperature-measurement points. Optimization shows that although the number of temperature-measurement points decreased by approximately 50%, the difference in MAE and MSE values were not significantly different. For four, five, and six measurement points, the MAE and MSE value decreased and the R value increased. This implies a higher prediction accuracy when setting the temperature measurement points at the optimal positions. The number of measurement points was optimized for cases with three, four, five, and six points, and the MSE values for these cases were 0.0415, 0.0234, 0.0189, and 0.0078, respectively. The MSE values tended to decrease when the number of measurement points increased; however, the values in each case were close to 0. Consequently, even if the number of measurement points was reduced to three, the prediction accuracy of ANNs can be expected to be very high if the measured temperature at the optimized position was used as the input value.

Table 10 shows the ANN test results when the sensors were optimally positioned. Figure 14 shows the comparison of the internal silo material levels obtained through experiments with the levels predicted by the ANN at the optimized position, showing the error between the two results. Figure 15 shows the comparison of the level-value errors predicted from the optimization positions via boxplot. As shown in Figure 15, the prediction error for 4, 5, and 6 points was approximately 50% lower than the prediction error for 3 points. As with the training results, the MAE and MSE values decreased by approximately 50%, and the R values increased. This confirms that if the number of measurement points was four, five, and six, the accuracy of the prediction results obtained based on the optimized input values was high.

## 6. Conclusions

Herein, we propose the use of an ANN to predict the level (output) of a material inside a silo based on the temperature (input) within the silo. The test data were experimentally obtained. The proposed method could accurately predict the material level of large-capacity silos, which was relatively difficult to measure in practice. A method to optimize the number and positions of temperature-measurement points required to predict material-level values was also proposed. The global search function of the GA was used to optimize the number and positions of the temperature measurement points because the accuracy of the ANN’s material-level prediction varied depending on the number and position of the temperature measurement points. The results indicated the following:The accuracy of the material levels predicted using temperature data was sufficiently high.The proposed method enables simultaneous, real-time monitoring of temperature and material levels using a temperature detector, thereby ensuring efficient silo management.The method can accurately predict the material levels inside a silo by optimizing the number and position of the temperature-measurement points. Even when the number of temperature measurement sensors was reduced from seven to three, the material level could be predicted accurately provided that the sensors were installed at optimized positions.When there are more than four measurement points, the error is 1.2–1.3%. This represents a 50% decrease from the error when the number of measurement points is three. Therefore, considering the economic feasibility and temperature detection performance, which is the existing function of the detector, it is considered optimal if the number of measurement points is four or five.The prediction error of the proposed method was approximately 50% less than that of the existing methods.The proposed method is expected to increase the efficiency of the silo operation, make it more economical, and improve the silo safety management in practical applications.

The limitation of this study is that we did not consider surface features when predicting the internal silo material-level values. Depending on the surface shape, the predicted level values and ground truth values may be larger than the conventional errors. While some data have shown information about such surface features, data on real surface features have been insufficient to perform ANNs. Therefore, we would like to obtain sufficient surface-shape data for future studies and check the material-level forecast considering this. In addition, the data used in this study were obtained for the colder months of October–December; hence, it is necessary to obtain and compare the data in the summer, which provides the opposite environment.

Moreover, we intend to check the predictability of factors affecting silo safety management (such as pressure acting on silo walls and the detector) based on temperature data obtained using a silo hotspot detector.

## Figures and Tables

**Figure 1 sensors-21-04427-f001:**
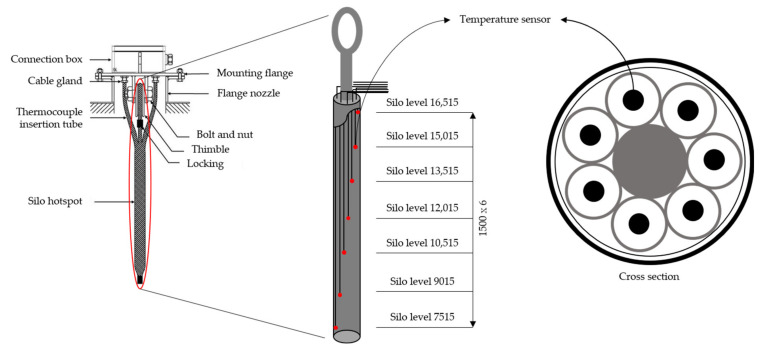
Silo hotspot detector (unit: mm).

**Figure 2 sensors-21-04427-f002:**
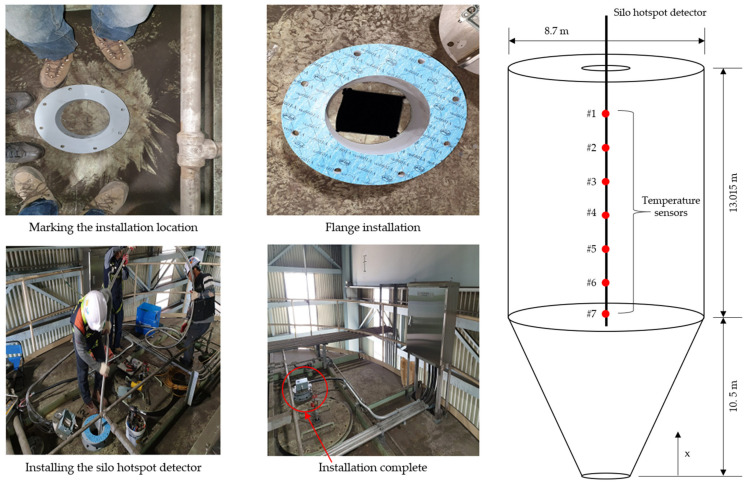
Installation of the silo hotspot detector and position of the temperature sensors.

**Figure 3 sensors-21-04427-f003:**
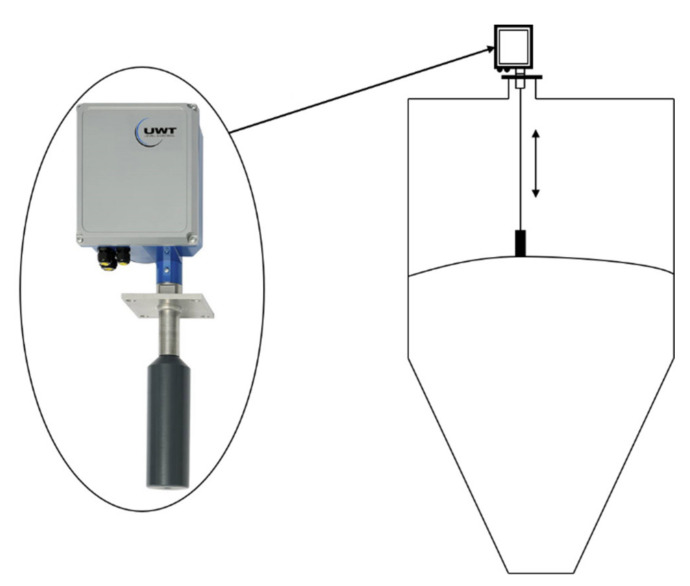
Material-level measuring device and method.

**Figure 4 sensors-21-04427-f004:**
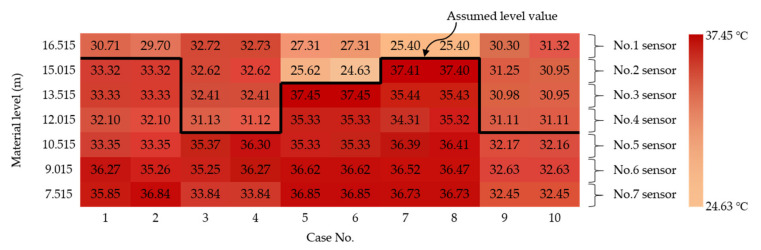
Experimental temperature results and assumed material-level values.

**Figure 5 sensors-21-04427-f005:**
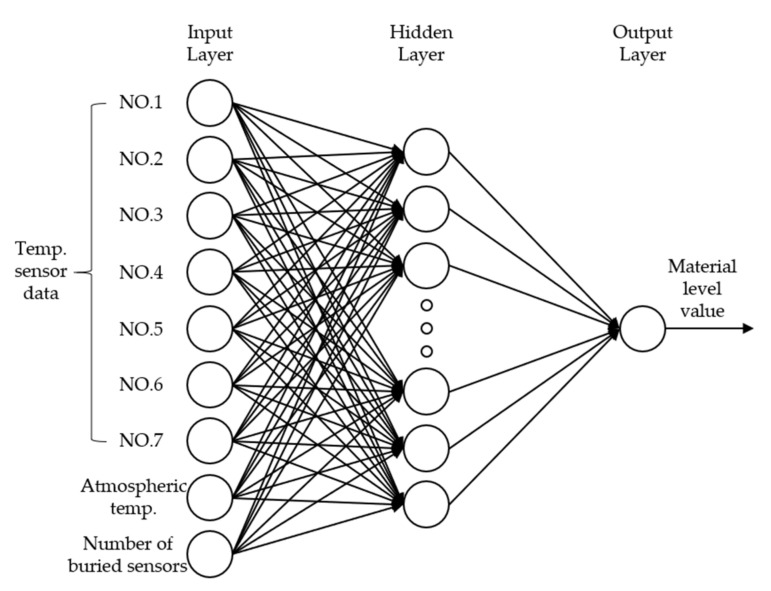
MFNN structure.

**Figure 6 sensors-21-04427-f006:**
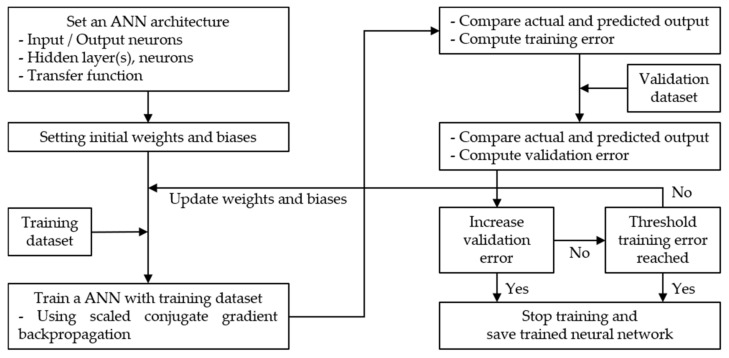
Flowchart of ANN training process.

**Figure 7 sensors-21-04427-f007:**
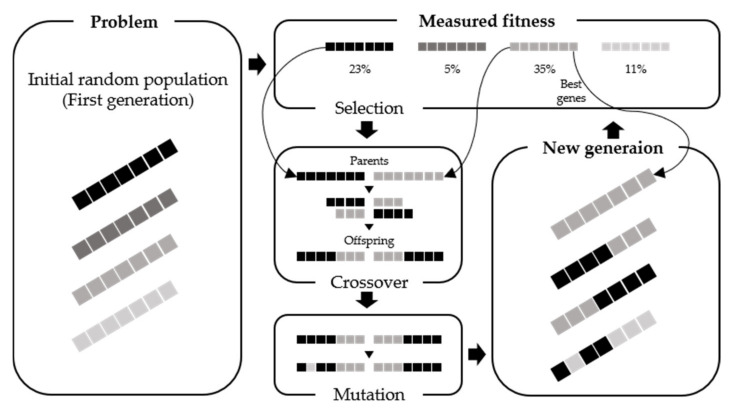
Illustration of the genetic algorithm concept, using an example iteration for a population of four individuals, each comprising seven genes.

**Figure 8 sensors-21-04427-f008:**
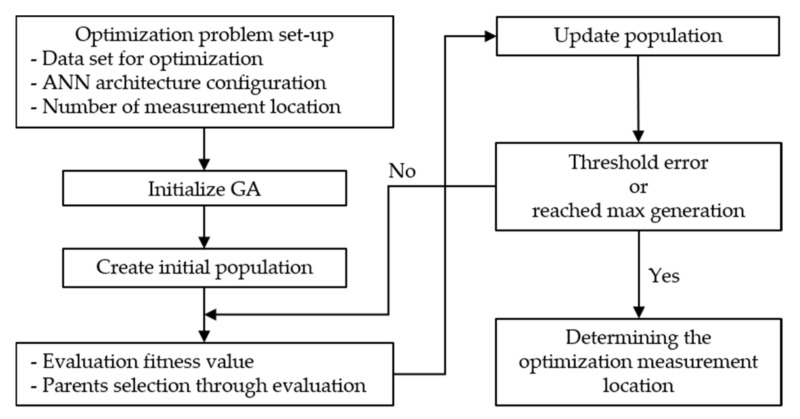
Flowchart of the ANN-GA.

**Figure 9 sensors-21-04427-f009:**
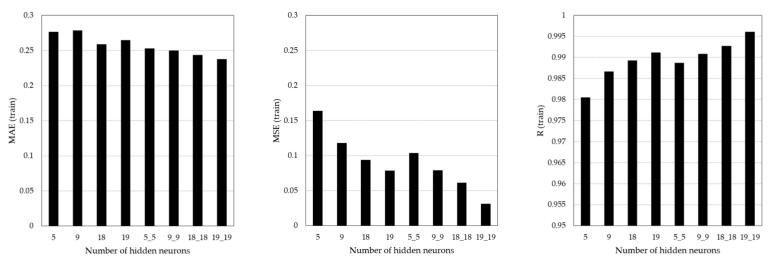
ANN training results.

**Figure 10 sensors-21-04427-f010:**
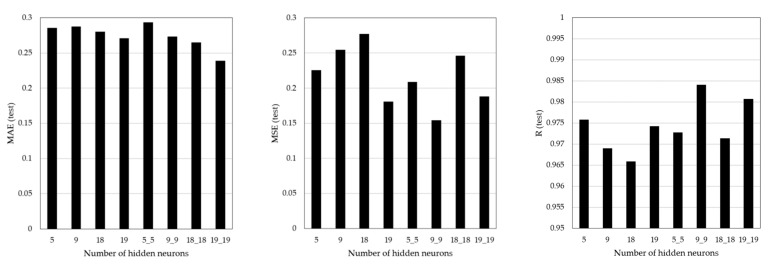
ANN test results.

**Figure 11 sensors-21-04427-f011:**
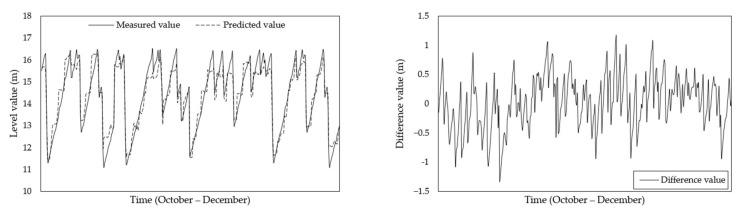
Level values and difference between measured and predicted values (Case 8: 9-19-19-1).

**Figure 12 sensors-21-04427-f012:**
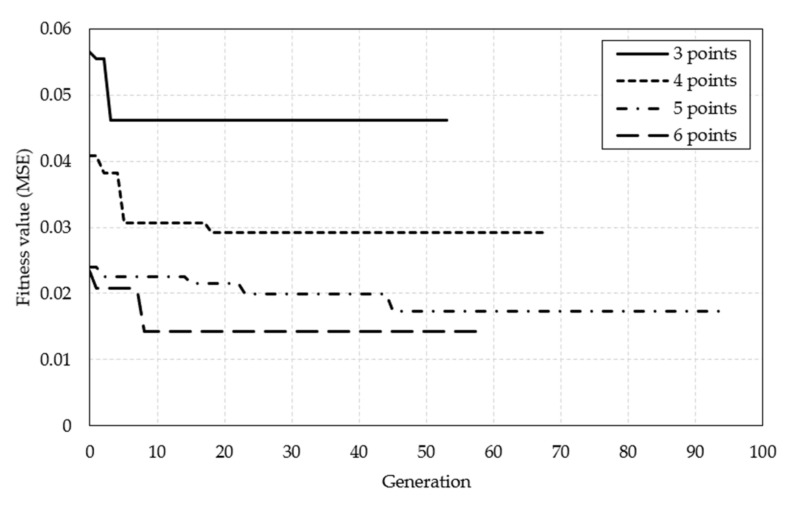
Best fitness and generation values based on the number of optimization points.

**Figure 13 sensors-21-04427-f013:**
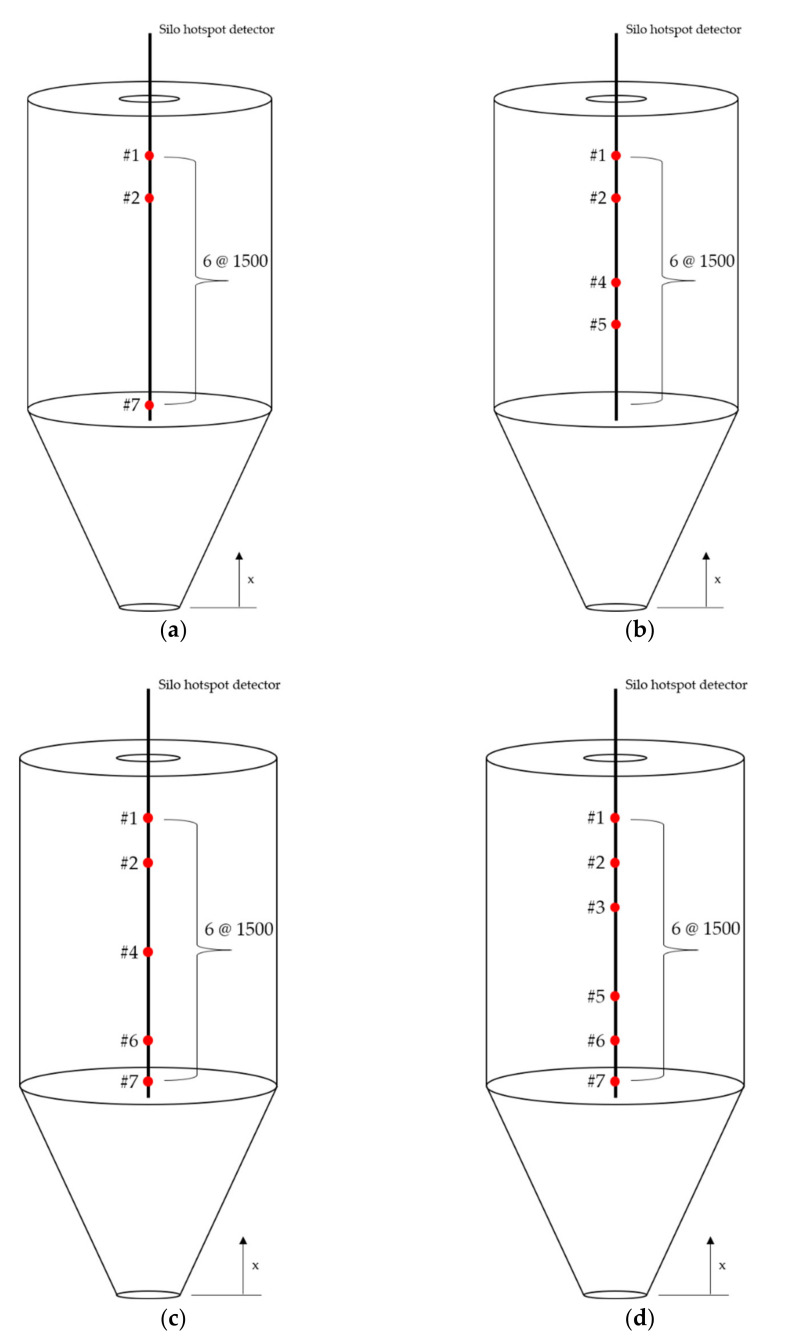
Measurement position according to the number of optimized temperature measurement points (unit: mm): (**a**) 3, (**b**) 4, (**c**) 5, and (**d**) 6 points.

**Figure 14 sensors-21-04427-f014:**
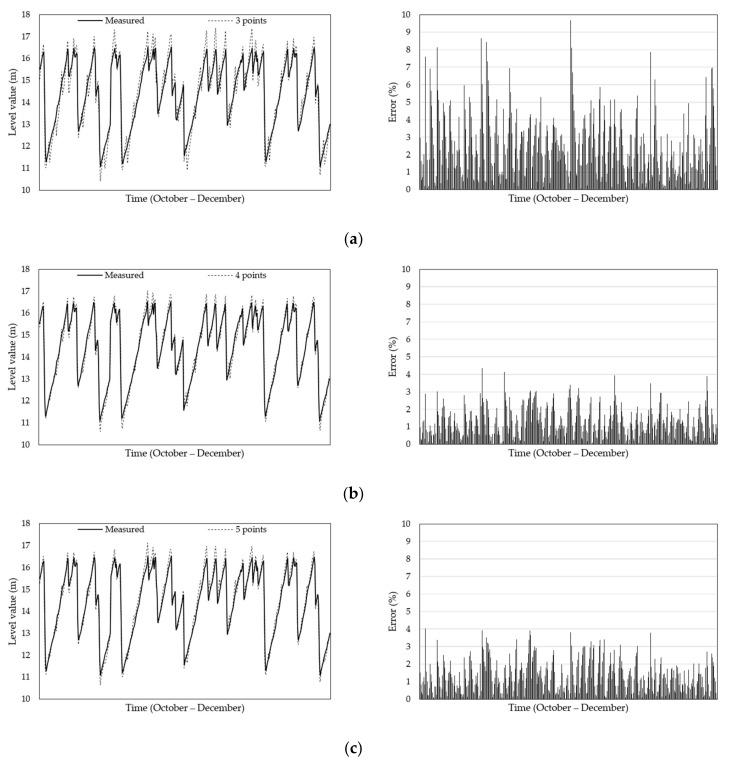
Comparison of predicted and measured material levels and error at optimization position. Results of (**a**) 3, (**b**) 4, (**c**) 5, and (**d**) 6 points.

**Figure 15 sensors-21-04427-f015:**
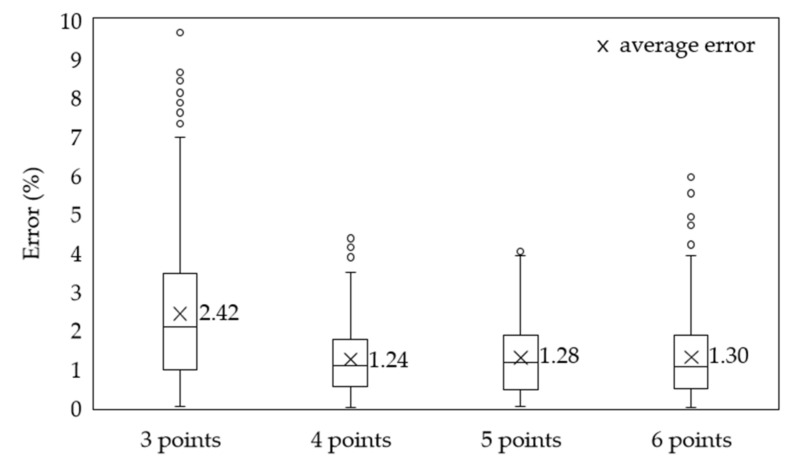
Comparison of error at optimization position.

**Table 1 sensors-21-04427-t001:** Specifications of the silo hotspot detector.

Parameters	Basic Values
Mounting type	Hanging down
Tensile load (kN)	126
Operating temperature (°C)	0–150
Temperature sensor	K-type, 1.6 Φ, ungrounded
Unit material	Stainless steel
Unit weight (kg/m)	3
Length (mm)	16,000
Diameter (mm)	≤35

**Table 2 sensors-21-04427-t002:** Specifications of the material level measuring tool.

Parameters	Basic Values
Measurement principle	Electromechanical lot-sensor
Version	Rope version
Process temperature (°C)	−40–80
Accuracy	1.5% of max. range
Min. immersion length (mm)	245
Min. immersion length (m)	1265

**Table 3 sensors-21-04427-t003:** The difference between the assumed level value and the measured value.

Case No.	1	2	3	4	5	6	7	8	9	10
Assumed value (m)	15.765	15.765	11.265	11.265	14.265	14.265	15.765	15.765	11.265	11.265
Measured value (m)	16.278	16.44	15.94	16.083	12.248	12.381	14.062	14.2	15.382	15.552
Difference value (m)	0.513	0.675	4.675	4.818	2.017	1.884	1.703	1.565	4.117	4.287

**Table 4 sensors-21-04427-t004:** Numbers of internal sensors at different material levels.

Level (m)	Number of Buried Sensors (ea)	Level (m)	Number of Buried Sensors (ea)
0–7.515	0	12.015–13.515	4
7.515–9.015	1	13.515–15.015	5
9.015–10.515	2	15.015–16.515	6
10.515–12.015	3	16.515–23.515	7

**Table 5 sensors-21-04427-t005:** Range of temperature and silo level values.

Data Type		Training Data		Test Data	
		Min.	Max.	Min.	Max.
Input	No. 1 (°C)	25	41.5	25.3	39.6
	No. 2 (°C)	22.9	41.8	24.7	39.8
	No. 3 (°C)	22.5	42.1	24.2	40.4
	No. 4 (°C)	29.5	42.4	31.9	40.2
	No. 5 (°C)	29.5	42.7	33	40.6
	No. 6 (°C)	29.3	45	31	41
	No. 7 (°C)	29	44.5	31.5	43.6
	Atmospheric temperature (°C)	16.5	28.1	16.5	20.1
	Number of buried sensors (ea)	2	7	3	7
Output	Material level (m)	9.83	19.89	11.11	19.76

**Table 6 sensors-21-04427-t006:** ANN model parameters.

Parameters	Basic Values
Number of input neurons	9
Number of hidden layers	1 or 2
Number of hidden neurons	5, 9, 18, or 19
Number of output neurons	1
Number of training dataset	3080
Number of test dataset	3080
Training algorithm	Scaled conjugate gradient
Transfer function	Logsig (hidden), purelin (output)
Learning rate	0.01
Momentum	0.9

**Table 7 sensors-21-04427-t007:** Parameters applied to ANN-GA model.

Parameters	Basic Values
Number of input neurons	*ts* (= 3–6)
Number of hidden layers	2
Number of hidden neurons	*Ts*
Number of output neurons	1
G_en_	150
N_pop_	10 × *ts*

**Table 8 sensors-21-04427-t008:** Model sensitivity according to input data combination.

Case No.	Feature Combination	Performance		
		MAE	MSE	R
1	F_1_	0.4634	0.1156	0.96843
2	F_1_ + F_2_	0.4266	0.0861	0.9789
3	F_1_ + F_3_	0.2602	0.0531	0.98738
4	F_1_ + F_2_ + F_3_	0.2528	0.0371	0.99107

**Table 9 sensors-21-04427-t009:** Training and Test results.

Case	Number of Hidden Layers	Structure	Performance					
			Training			Test		
			MAE	MSE	R	MAE	MSE	R
1	1	9-5-1	0.2768	0.164	0.98055	0.2859	0.2258	0.9758
2		9-9-1	0.2786	0.1179	0.98669	0.2878	0.2545	0.96896
3		9-18-1	0.2587	0.0942	0.9893	0.2805	0.2771	0.96589
4		9-19-1	0.2646	0.0786	0.9912	0.2709	0.1808	0.97421
5	2	9-5-5-1	0.2528	0.1037	0.98875	0.2934	0.2088	0.97274
6		9-9-9-1	0.2501	0.0794	0.99082	0.2735	0.1545	0.98406
7		9-18-18-1	0.2434	0.0615	0.99277	0.2652	0.2463	0.97135
8		9-19-19-1	0.2377	0.0315	0.99612	0.2391	0.188	0.9807

**Table 10 sensors-21-04427-t010:** ANN training and test result at optimized temperature-measurement position.

Number of Measurement Points	Structure	Performance					
		Training			Test		
		MAE	MSE	R	MAE	MSE	R
3	5-11-11-1	0.3384	0.0415	0.98939	0.408	0.2016	0.97290
4	6-13-13-1	0.1772	0.0234	0.99463	0.2152	0.1087	0.97399
5	7-15-15-1	0.1707	0.0189	0.99561	0.1878	0.1071	0.97771
6	8-17-17-1	0.141	0.0078	0.99795	0.1541	0.0919	0.98393

## Data Availability

Not applicable.

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
