# Peer review of "Optimization of Position and Number of Hotspot Detectors Using Artificial Neural Network and Genetic Algorithm to Estimate Material Levels Inside a Silo"

_sensors, 2021, doi:10.3390/s21134427_

Round 1

Reviewer 1 Report

There is not the state of the art review. I do not know how the real contribution of this work is. It is mandatory the authors insert the section about the state of the art of the problem.

Please insert the contributions of the work.

It is not clear the real benefits associated with GA. The authors need to explain more about GA utilization. Why was the number of hidden neurons tested to 5, 9, 18, or 19? Will it work with 100 neurons?

Please, remove this affirmation: “ANNs can be classified into multi-layer perceptrons (MLPs), radial basis functions,  wavelet neural networks, self-organizing maps, and current networks, depending on the order and information-processing method of the connected neurons”

Equations 1 and 2 are lost. They never were cited in the text.

Fix paragraph in Line 242

Reviewer 2 Report

The submitted manuscript tackles the problem of precise determination of a silo contents level based on the measurements gathered from temperature sensors embedded in the silo. The authors propose for this purpose a specific architecture of a (shallow) feed-forward neural-network taught on a set of experimental data. Additionally, the authors employ a genetic algorithm to optimize the number and position of the temperature sensors in such a way that the requested prediction accuracy can be obtained from minimal number of sensors.

The problem is clearly stated, the proposed solutions are sound and precisely described, and finally the presented results strongly support the approach chosen by the authors.

All things considered I find the submitted manuscript to be a valuable piece of engineering work which is interesting both as a specific application and as an inspiration for quite wide range of similar problems.

To conclude, I support publication of the paper.

Author Response

Response to Reviewer 2 Comments

Dear Reviewers:

Thank you for your letter and for the reviewer's comments concerning our manuscript entitled “Optimization of Position and Number of Hotspot Detectors using Artificial Neural Network and Genetic Algorithm to Estimate Material Levels Inside a Silo.” (ID: sensors-1258057).

We appreciate for Reviewers’ warm work earnestly.

Once again, thank you very much for your comments.

Thank you and best regards.

Yours sincerely,

Jeong Hoon Rhee

Reviewer 3 Report

The authors of the paper describe their proposed approach Optimization of Position and Number of Hotspot Detectors using Artificial Neural Network and Genetic Algorithm to Estimate Material Levels Inside a Silo. The topic is interesting and with possible applicability. However, the paper needs several improvements:

1) the main contribution and originality should be explained in more detail, is GA applied to NNs?

2) the motivation of the approach needs further clarification, why GAs and not other metaheuristic?

3) discussion of related work in GA for ANNs should be expanded with more recent work

4) Minor grammar and syntax issues need correction

5) more simulation results and formal comparison of results are needed

6) the conclusions should be extended with more future work

7) More references to GA for NNs papers should be included, like:

Comparison of particle swarm optimization variants with fuzzy dynamic parameter adaptation for modular granular neural networks for human recognition. J. Intell. Fuzzy Syst. 38(3): 3229-3252 (2020)

Multi-objective optimization for modular granular neural networks applied to pattern recognition. Inf. Sci. 460-461: 594-610 (2018)

A Grey Wolf Optimizer for Modular Granular Neural Networks for Human Recognition. Comput. Intell. Neurosci. 2017: 4180510:1-4180510:26 (2017)

Optimization of modular granular neural networks using a firefly algorithm for human recognition. Eng. Appl. Artif. Intell. 64: 172-186 (2017)

Reviewer 4 Report

I have several comments as follows:

The sensitivity of the model with different combination of the input should be investigated.

Other performance indexes other than MSE and R should be included.

The training and testing results should be combined in one table.

Round 2

Reviewer 1 Report

The authors have made all suggestions, and they inserted new information to clarify the manuscript.

Reviewer 3 Report

The authors have addressed all my concerns and the paper can be accepted.